# Land-Cover Classification Using MaxEnt: Can We Trust in Model Quality Metrics for Estimating Classification Accuracy?

**DOI:** 10.3390/e22030342

**Published:** 2020-03-17

**Authors:** Narkis S. Morales, Ignacio C. Fernández

**Affiliations:** Centro de Modelación y Monitoreo de Ecosistemas, Facultad de Ciencias, Universidad Mayor, Santiago 8340589, Chile

**Keywords:** Akaike information criterion (AIC), area under the curve (AUC), bayesian information criteria (BIC), classification accuracy, Kappa, land-cover, model quality, one-class classification

## Abstract

MaxEnt is a popular maximum entropy-based algorithm originally developed for modelling species distribution, but increasingly used for land-cover classification. In this article, we used MaxEnt as a single-class land-cover classification and explored if recommended procedures for generating high-quality species distribution models also apply for generating high-accuracy land-cover classification. We used remote sensing imagery and randomly selected ground-true points for four types of land covers (built, grass, deciduous, evergreen) to generate 1980 classification maps using MaxEnt. We calculated different accuracy discrimination and quality model metrics to determine if these metrics were suitable proxies for estimating the accuracy of land-cover classification outcomes. Correlation analysis between model quality metrics showed consistent patterns for the relationships between metrics, but not for all land-covers. Relationship between model quality metrics and land-cover classification accuracy were land-cover-dependent. While for built cover there was no consistent patterns of correlations for any quality metrics; for grass, evergreen and deciduous, there was a consistent association between quality metrics and classification accuracy. We recommend evaluating the accuracy of land-cover classification results by using proper discrimination accuracy coefficients (e.g., Kappa, Overall Accuracy), and not placing all the confidence in model’s quality metrics as a reliable indicator of land-cover classification results.

## 1. Introduction

Maximum entropy-based algorithms have been widely used in biological sciences, such as biochemistry, physiology, neurobiology, genetics and animal behavior [1], and have gained a predominant role as methodological approaches for species distribution modelling in biogeographical disciplines [2,3]. One of the most popular maximum entropy-based software used for species distribution modeling is MaxEnt [4,5]. The popularity of MaxEnt is probably due to its free availability, good or better performance than alternative modeling methods, ability to rely on presence data only, an intuitive visual interface and the availability of default parameter setting that facilitates it use by new users. Nevertheless, these characteristics have also tempted researchers to use MaxEnt as a black box, frequently generating non-optimal models that can be reporting sub- or over-estimated species distribution outcomes [6].

While MaxEnt was originally developed as a species distribution modeling software, its built-in algorithm can be applied for other spatial modeling tasks, being increasingly used as a single-class land-cover classification. For example, MaxEnt has been used for mapping urban land uses in California [7], urban land in China [8], urban vegetation types in Chile [9], invasive plant species in Colorado [10] and California [11], conservation habitats in Germany [12] and habitat loss and land-cover classes in Italy [13,14]. In comparison to commonly used land-cover classification methods, such as the Maximum Likelihood Classification (MLC), single-class classification methods like MaxEnt have the advantage of focusing on the land-covers of interest, avoiding using resources and efforts in classifying land-covers that are not relevant for the study objectives [15,16].

However, because MaxEnt was originally designed to be used for species distribution modeling, and not for land-cover classification, recommended procedures for generating high-quality species distribution models (e.g., [3,4,6]) may not hold for providing accurate results for land-cover classification [16]. One of the main factors related to this issue is the differences in the methodological approaches used for measuring the accuracy of the generated modeling outcomes, which for species distribution modeling is often estimated based on continuous probabilistic data, while for land-cover classification is based on binary presence/absence data. These differences are tightly related with the step at which the accuracy of the modeling output is measured, i.e., before transforming the probabilistic output into binary maps for species distribution modeling, and after performing this step for land-cover classification.

MaxEnt comes preconfigured to use default parameter settings (e.g., auto features), however, the user can modify several parameters, providing the possibility to generate a large number of models [5]. Therefore, the modeling outcome can be based in any of these potential models. Because the large number of potential models in species distribution modeling selecting the “best model” is a key step for ensuring the accuracy of the modeling outcome. There are several model quality metrics that can be used, including the Area Under the Curve (AUC), Akaike Information Criterion (AIC), Akaike Information Criterion corrected for small sample sizes (AICc) and Bayesian information Criteria (BIC) [2,3]. These quality metrics are applied to the probabilistic model generated by MaxEnt, but not to the presence/absence maps generated after applying binary threshold values to the probabilistic output [17].

On the other hand, in the case of land-cover classification accuracy, quality metrics are commonly computed after transforming the probabilistic models into a binary presence/absence land-cover maps using classification accuracy metric such as the Kappa coefficient and the Overall Accuracy (e.g., [16,18,19]). MaxEnt provides a set of 11 different thresholds values for generating the binary maps, but researchers can technically use any threshold values ranging from 0 to 1 (e.g., [7]). A recent study showed that while the model parameterization could have a large impact on classification results depending on the land-cover, the most relevant factor related to the accuracy of land-cover classification is the selected threshold [16]. This may imply that placing efforts in finding the best parameters for generating high-quality probabilistic models may not be necessary for land-cover classification using MaxEnt, but also, that reporting land-cover classification accuracies based on MaxEnt model quality metrics (e.g., AUC) may be largely inaccurate for binary maps.

Nevertheless, to the best of our knowledge, currently there is no information on how the quality of MaxEnt probabilistic models relate to the accuracy of the final land-cover classification binary maps, and if these relationships (if any) are influenced by the threshold used and land-cover under analysis. Thus, in this work we aimed to evaluate if commonly used model quality metrics, such as AUC, AIC and BIC, are suitable proxies for estimating the precision of land-cover classification outcomes produced through MaxEnt. We also evaluate if these results depend on the threshold values used for generating the binary maps and the land-cover being classified.

## 2. Materials and Methods

### 2.1. Study Area

The research was carried out using the metropolitan area of Santiago de Chile as a case study (33.4489° S, 70.6693° W). This is the capital city of Chile and is estimated to harbor a population of 5.3 million inhabitants within an urban area of 875 km^2^ [16]. Santiago has a Mediterranean climate, with cold and rainy winters and dry and hot summers. The seasonal variations in precipitations and temperatures between winter and summer months generate distinctive phenological changes in Santiago´s vegetation (See Appendix A), allowing to discriminate between different vegetation functional types from satellite images.

### 2.2. Remote Sensing Imagery

The land-cover classification process was performed using a set of remote sensing images acquired from the Sentinel-2 mission satellites. We selected cloud-free images, representing the vegetation conditions of summer (6 March 2016) and winter (2 August 2016) for Santiago city. From the 13 bands provided by the Sentinel-2 satellites for each date, we only used the bands with native resolution of 10m/pixel (i.e., red, green, blue, Near Infrared (NIR)). From these bands we derived the Normalized Difference Vegetation Index (NDVI), and calculated an additional layer representing the arithmetic difference between summer and winter NDVI. After this process we ended up with 11 layers, five for each season plus the one representing seasonal NDVI differences. All the layers were processed using QGIS 3.82 Zanzibar (http://www.qgis.org).

### 2.3. Selection of Representative Land-Cover Points

We used very-high resolution images (<1 m/pixel) available through Google Earth 7.3.2 to manually select 100 ground-true points for four types of land covers: (1) Built-up, which included houses, buildings, commercial areas and paved surfaces; (2) Grass, related to any surface covered by green lawns all-year around; (3) Evergreen trees, corresponding to trees and shrubs that retain their leaves active throughout the year; (4) Deciduous trees, associated to trees and shrubs that lose their leaves during winter months. The images used for this task corresponded to the same seasons and year of the Sentinel-2 images. We double checked each of the selected points to correct any potential mismatch between the Google Earth and Sentinel images due to resolution discrepancies.

### 2.4. Land-Cover Classification

We used the maximum entropy algorithm available through MaxEnt software 3.4.1 (https://biodiversityinformatics.amnh.org/open_source/maxent), as a one-class classification tool for discriminating between the four land-cover types used for the analysis. To avoid potential issues with over-parameterization, a Pearson correlation analysis of the 11 Sentinel-2 selected layers was performed, keeping only those layers having correlations smaller than 0.8. After this analysis we ended up with seven layers: Red (summer/winter), NIR (summer/winter), NDVI (summer/winter) and the seasonal difference of NDVI. We used a set of 45 different computational combinations of MaxEnt “feature classes” and “regularization multipliers” following Morales et al. [6]. The features used were lineal (L), hinge (H), quadratic (Q), threshold (T) and product (P); and the regularization multipliers were: 0.25, 1, 3 and 5. Each combination was replicated five times for each land-cover. In total, we generated a set of 900 classification maps (45 parameters combinations × 4 land-covers types × 5 replicates). We used the average of the 5 replicates as the model outcome per combination of parameters, therefore generating 45 maps per land-cover type. To generate the final binary maps, we applied each of the 11 binary thresholds provided by MaxEnt to each of the 45 maps per land-cover. The binary thresholds used were: cumulative value 1 threshold (FC1); fixed cumulative value 5 threshold (FC5); fixed cumulative value 10 threshold (FC10); 10 percentile training presence threshold (10PTP); balance training omission, predicted area and threshold value threshold (BTOPA); equate entropy of thresholded (EETD) and original distributions threshold (EETOD); equal test sensitivity and specificity threshold (ETeSS); equal training sensitivity and specificity threshold (ETrSS); maximum test sensitivity plus specificity threshold (MTeSPS); minimum training presence threshold (MTP); maximum training sensitivity plus specificity threshold (MTrSPS). After applying the thresholds, we generated a total of 1980 classified maps (see Appendix A). Application of thresholds for generating the binary classified maps were performed using R version 3.5.1 ([20]).

### 2.5. Classification Accuracy Metrics

We evaluated the accuracy of the classification result by estimating two accuracy metrics, the Kappa coefficient and the Overall Accuracy (OA). To calculate these metrics, we randomly sampled 1000 points within 16 × 16 km quadrant, representing the largest square fitting the irregular shape of Santiago. These points were visually inspected through Google Earth, and classified into built-up, grass, evergreen and deciduous trees. All points falling on places where more than one cover-class was present on a 5 m radius were moved to the closest area where the land-cover within that radius coincided with the dominant on the original location. These 1000 points were used to build a testing layer for each land-cover class based on positive and negative labels. Then, we used the testing layers to calculate the Kappa and OA for each classification result following the formulas shown in Fielding and Bell [21].

### 2.6. Model’s Quality Metrics

We used the AUC_Training_, AUC_Testing_, AIC, AICc and BIC as the model´s quality metrics. AUC_Training_ and AUC_Testing_ were obtained directly from MaxEnt outputs. To calculate AIC, AICc and BIC statistics we used the ENMTOOLS software version 1.4.4 [22]. To calculate AIC, AICc and BIC statistics we used the ENMTOOLS software version 1.4.4 [22] and the outputs from the land-cover models from MaxEnt [22] and the outputs from the land-cover models from MaxEnt (see Appendix A). All these metrics were calculated based on the averaged values of the five replicates generated for each model (See Section 2.4).

### 2.7. Relationships beween Model’s Quality and Land-Cover Classification Accuracy

We evaluated the relationships between the five model´s quality and the two land-cover classification accuracy metrics by performing a series of Spearman correlation analysis. We decided to use this non-parametric test because most of the metrics showed non-normal distribution (See Appendix A). We first evaluated the relationships among the model´s quality metrics, and then evaluated the relationships between the model´s quality and the land-cover classification accuracy for each of the thresholds used for building the binary maps. All the statistical analyses were performed using R 3.6.1 [20].

## 3. Results

### 3.1. Relationship between Model’s Quality Metrics

Correlation analysis between the model quality metrics tend to show consistent patterns for the relationships between metrics, but not for all land-covers (Figure 1). For example, AUC_Testing_ and AUC_Training_ showed a positive correlation between them for all land-covers, except for grass where no correlation is present. Also, AUC_Training_ showed a negative correlation with AIC, AICc and BIC for all land covers, but not for built. AUC_Testing_ was the metric showing the less stable pattern, with no relationship with AIC, AICc and BIC for built nor grass, but with negative relationship with these metrics for evergreen and deciduous cover. On the other hand, AIC, AICc and BIC showed consistent patterns of associations between them, presenting very strong positive correlations for all the analyzed land-covers (Figure 1).

### 3.2. Relationship between Model Quality and Classifiaction Accuracy

Correlation analysis between model quality and classification accuracy metrics show that there is a relationship between the statistical quality of the model and the accuracy of classification results, but the degree of this association largely depends on the land-cover under analysis (Figure 2). For example, while for built cover there seems to be no consistent relationships between model quality and classification accuracy, for grass, evergreen and deciduous cover there are some clear and consistent association patterns. For the three vegetation covers, AUC_Training_ tend to be strongly positively correlated with classification accuracy, whereas AIC, AICc and BIC negatively correlated to the classification metrics. Among this, AIC is the model quality metric presenting the strongest relationships with classification accuracy. AUC_Testing_ is the only quality metric showing contrasting results between vegetation covers. While for grass, AUC_Testing_ shows no relationship with land-cover classification accuracy, for evergreen and deciduous cover shows a positive relationship. These general patterns hold independent of the classification accuracy metric used for the analysis, i.e., Kappa or overall accuracy (Figure 2).

The thresholds used for generating the binary maps also have an impact on the magnitude of the relationship between the model’s quality and classification accuracy metrics (Figure 2). Nevertheless, there are no clear patterns that may shed some light on what thresholds provide better classification results. For built cover, the thresholds seem to have a random effect on classification results, even showing opposite correlation coefficients when some thresholds are applied (e.g., FC10, MTP). For the vegetation covers, the selection of thresholds does change the relationship between the model´s quality and classification results, however no threshold shows to ensure a fit between the quality of the model and the final classification, as this greatly differ among vegetation cover and the model´s quality metrics (Figure 2).

## 4. Discussion

An increasing number of studies have been using and testing the capacity of the built-in maximum entropy algorithm available in MaxEnt as a tool for performing land-use and land-cover classification based on remote sensing imagery (e.g., [7,15,16,23]). However, as MaxEnt was originally developed as a niche modelling tool, but not a land cover classification tool, there still are several gaps of information on how to configure the built-in parameters and specific thresholds to obtain best classification results [7,11,16,18,19]. In fact, to the best of our knowledge, until this study, there was no information on how the quality of MaxEnt’s probabilistic models may affect the accuracy of binary land-cover classification.

The performance of MaxEnt models are usually assessed based on AUC values [24], but other model quality metrics based on parsimony have also been suggested (e.g., [2,25]). Evaluating the quality of the model is crucial for relating these results to classification accuracy. However, as our results show, choosing one quality metric instead of another could bias the analysis, as some metrics present inconsistent relationships between them depending on the analyzed land cover. For example, AUC_testing_ had a positive correlation with AUC_training_ for all land-covers except for grass, no correlation with AIC, AICc and BIC for built and grass, negative correlation with these metrics for evergreen, but only with AICc and BIC for deciduous. On the other hand, AIC, AICc and BIC were consistently strongly positively correlated between them, with very similar patterns for all analyzed land-covers. In this regard, the variability we found on AUC metrics supports previous studies suggesting not to rely on AUC metrics for analyzing MaxEnt model´s quality [2,6,24,26].

Regarding the relationship between the model´s quality metrics and land-cover classification accuracy, we found that none of the tested quality metrics provided a consistent association with the accuracy of classification results for all land covers and thresholds. While for built cover, there was no consistent patterns of correlations for any quality metrics, for grass, evergreen and deciduous, AUC_training_, AIC, AICc and BIC tended to show consistent associations with classification accuracy. This result suggests that the quality of the model can have an effect on the accuracy of the resulting land-cover classifications, but this effect could largely differ depending on the land-cover under analysis, the threshold used for building the binary maps and the metric used for evaluating the quality of the model [16]. Therefore, while for some cases the quality of the model has a positive effect on the classification accuracy (e.g., [27]), in other situations, there may be no relation between them (e.g., [28]).

Selection of thresholds for generating the binary maps has been identified as one of the most important factors for producing accurate classification results with MaxEnt [7,16]. Thresholds are applied once the probabilistic models are already built, so they have no effect on the model´s quality but may have a great influence on classification accuracy. What is interesting from our work is that thresholds do not seem to affect the general relationship between model quality and classification accuracy for grass, evergreen and deciduous covers (i.e., better models tend to produce better classification results). However, for built cover, thresholds do affect the relationship between model quality and classification accuracy.

We do not have a specific explanation for these findings, but based on Fernández and Morales [16], we think that this discrepancy could be related to the interaction between modeling parameters and the intrinsic spatial structure of the land-covers that are classified, particularly on how this interaction could increase or reduce the quality of the generated models. For example, discriminating among vegetation types (e.g., evergreen from all others) may require higher quality models capable to distinguish pixels with the target vegetation from other vegetated pixels with similar spectral signatures. Therefore, for these situations, achieving a high-quality model will have a large influence on classification accuracy, which will be translated to the final result independently of the threshold used. On the other hand, because built areas are more spectrally homogenous, classifying these areas may be feasible even with low quality models, therefore, MaxEnt will tend to generate models of lower quality whose potential for producing accurate classification results could be greatly affected by the selection of the binary threshold.

## 5. Conclusions

Results from our work suggest that while placing efforts in generating a model of higher quality could increase the chances of getting higher classification accuracies, there are other factors, such as the intrinsic spatial structure of the land-cover under analysis and the threshold used for building the binary maps, that could be of great relevance for achieving accurate classification results. In this regard, a rule of thumb is always evaluating the accuracy of classification results by using proper discrimination accuracy coefficients (e.g., Kappa, Overall Accuracy), and not placing all the confidence in the model’s quality metrics (e.g., AUC, AIC, BIC) as a reliable indicator of land-cover classification performance.

## Figures and Tables

**Figure 1 entropy-22-00342-f001:**
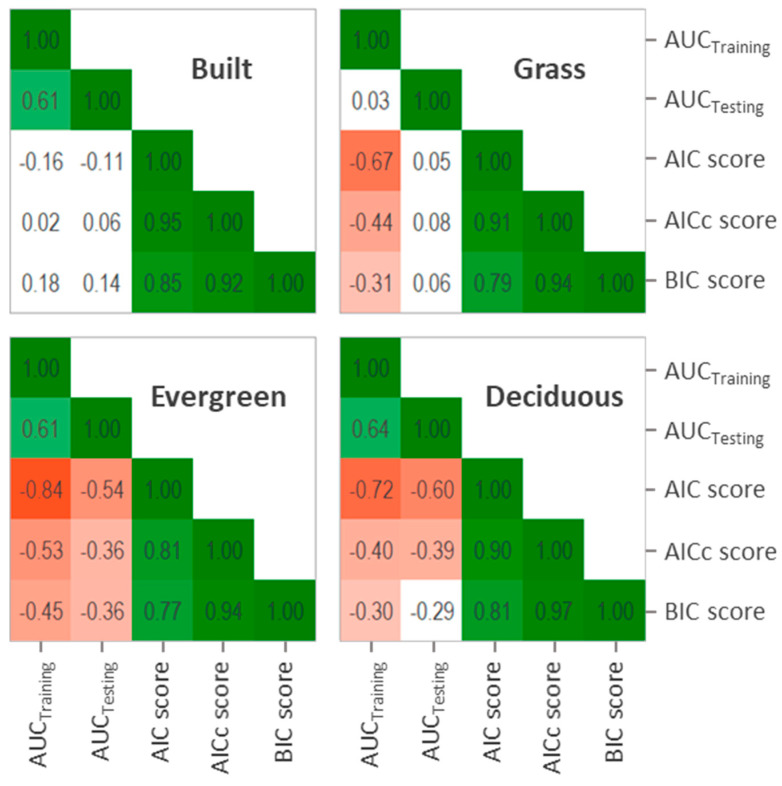
Correlation matrix between the five MaxEnt model’s quality metrics for the four analyzed land-covers. Each square shows the Spearman correlation coefficient (Rho) resulting from comparing 45 models (n = 45). Squares are colored based on Rho values from green (+1) to red (−1), except for relationships statistically not different from 0 at *p* < 0.05, which are shown in white. *p*-values for all correlations are shown in Appendix A.

**Figure 2 entropy-22-00342-f002:**
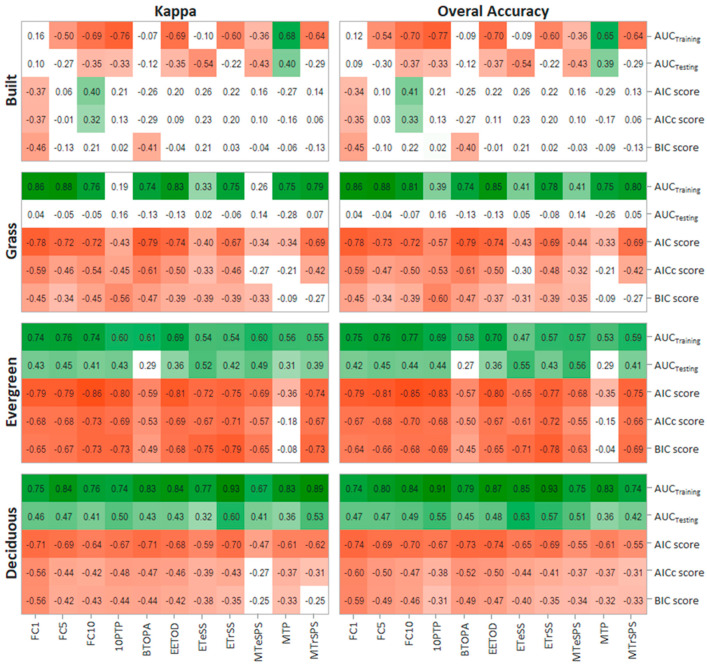
Correlation matrix between the five MaxEnt model’s quality metrics and the two classification accuracy metrics for the eleven thresholds used for building the binary maps and the four analyzed land-covers. Each square shows the Spearman correlation coefficient (Rho) resulting from comparing 45 models (n = 45). Squares are colored based on Rho values from green (+1) to red (−1), except for relationships statistically not different from 0 at *p* < 0.05, which are shown in white. *p*-values for all correlations are shown in Appendix A.

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
