# Peer review of "Land-Cover Classification Using MaxEnt: Can We Trust in Model Quality Metrics for Estimating Classification Accuracy?"

_entropy, 2020, doi:10.3390/e22030342_

Round 1
Reviewer 1 Report
COMMENTS
I do like the idea behind the manuscript, that is to use MaxEnt as a classification model for different land cover classes. However, I have several questions and suggestions related to the lack of clarity of what is truly calculated and how it is calculated, as well as the motivations behind the use of MaxEnt for the stated purpose.
As for the latter, my concern is about why one would use MaxEnt to re-classify something that has been already classified with quite a lot of expertise and/or minimum uncertainty about the classified elements, i.e. land cover. In other words, why somebody would you MaxEnt to classify land-cover that has been already classified?
Alternative uses of MaxEnt have been done for landslides and floods, for instance see Convertino et al., 2013 (J Geophy Res), or Servadio et al 2018 in which a more generalized MaxEnt model is provided to define systemic indicators of patterns given uncertainty and sensitivity defined by variable probability distribution functions. These studies as well as others make also use of already classified maps, such as land cover (considering their uncertainty) and therefore I do see why one would reclassify land cover maps that are complete. Uncertainty of classification that can also be quantified via global sensitivity and uncertainty analyses (e.g. Convertino et al, 2013, EnvSoft). Of course this leaves aside the case of when land cover maps are incomplete.
As for the second topic, i.e. how reclassification is calculated, I believe the authors must explain, also with a conceptual diagram, how different MaxEnt value ranges are corresponding to different land cover classes. What is the rationale for using these classes? And what is the uncertainty related in the selection of these thresholds?
RECCOMENDATION
For the above reasons I recommend Major Revisions. I believe these revision can clarify several aspects of the paper that are a bit confusing as of now. THX.
REFERENCES
Detecting fingerprints of landslide drivers: A MaxEnt model
M. Convertino A. Troccoli F. Catani
J Geophy Res, First published: 01 July 2013 https://doi.org/10.1002/jgrf.20099
Untangling drivers of species distributions: Global sensitivity and uncertainty analyses of MaxEnt
Matteo Convertino, R. Muñoz-Carpena, M. L. Chu-Agor, G. A. Kiker, I. Linkov
10.1016/j.envsoft.2013.10.001
Optimal information networks: Application for data-driven integrated health in populations
Joseph L. Servadio1 and Matteo Convertino2,3,4,*
Science Advances 02 Feb 2018:
Vol. 4, no. 2, e1701088
DOI: 10.1126/sciadv.1701088
Author Response
Reviewer 1
Comments:
I do like the idea behind the manuscript, that is to use MaxEnt as a classification model for different land cover classes. However, I have several questions and suggestions related to the lack of clarity of what is truly calculated and how it is calculated, as well as the motivations behind the use of MaxEnt for the stated purpose.
As for the latter, my concern is about why one would use MaxEnt to re-classify something that has been already classified with quite a lot of expertise and/or minimum uncertainty about the classified elements, i.e. land cover. In other words, why somebody would you MaxEnt to classify land-cover that has been already classified?
Alternative uses of MaxEnt have been done for landslides and floods, for instance see Convertino et al., 2013 (J Geophy Res), or Servadio et al 2018 in which a more generalized MaxEnt model is provided to define systemic indicators of patterns given uncertainty and sensitivity defined by variable probability distribution functions. These studies as well as others make also use of already classified maps, such as land cover (considering their uncertainty) and therefore I do see why one would reclassify land cover maps that are complete. Uncertainty of classification that can also be quantified via global sensitivity and uncertainty analyses (e.g. Convertino et al, 2013, EnvSoft). Of course this leaves aside the case of when land cover maps are incomplete.
Response: We think that the reviewer misunderstood the methodology we used in the paper. We did not used an already classified land-cover map for our exercise, instead we used a set of non-classified satellite images (Sentinel 2 images) as input material, and used MaxEnt as a tool for classifying this images into four different land cover types. To validate the classification results we used true-points obtained by visual interpretations of very-high resolution images available in Google Earth. Perhaps the reviewer assumed that the latter process was based on a previously generated land-cover map for the study area, but it was not the case.
As for the second topic, i.e. how reclassification is calculated, I believe the authors must explain, also with a conceptual diagram, how different MaxEnt value ranges are corresponding to different land cover classes. What is the rationale for using these classes? And what is the uncertainty related in the selection of these thresholds?
Response: We do not completely understand the point of the reviewer. We think that this comment is also based on a wrong interpretation of what we did in our study, as we did not “reclassify” any land-cover map. In relation to why we used these classes, we attempted to use different classes of vegetation to evaluate the capacity of this approach for discriminating between them based on their phenological changes, but also including a more seasonal-stable cover, represented here by built cover (Please see Fernández and Morales 2019 https://doi.org/10.7717/peerj.7016 for more details on this).
Reviewer 2 Report
Revision of the ms “Land-cover classification using MaxEnt: ¿Can we trust in model quality metrics for estimating classification accuracy?
Line 2 = delete the question mark upside down
Lines 10 – 11 = I prefer “…especially for species distribution modelling”
Lines 27 – 28: Place the keywords in alphabetic order
Line 34 = I prefer “…especially for species distribution modelling”
Line 36 = After “…distribution modeling is MaxEnt”, you should add also the following reference:
Phillips, S. J., Anderson, R. P., Dudík, M., Schapire, R. E., & Blair, M. E. (2017). Opening the black box: An open‐source release of Maxent. Ecography, 40(7), 887-893.
Line 43 = After “While MaxEnt was originally developed as a species distribution modeling software…”, you should add also the following recent references as examples:
Bosso, L., Ancillotto, L., Smeraldo, S., D’Arco, S., Migliozzi, A., Conti, P., & Russo, D. (2018). Loss of potential bat habitat following a severe wildfire: a model-based rapid assessment. International Journal of Wildland Fire, 27(11), 756-769.
Line 47 = You have written two times “and California”.
Lines 71 – 72 = After “These quality metrics are applied to the probabilistic model generated by MaxEnt…”, you should add also the following recent references as examples:
Tanner, E. P., Orange, J. P., Davis, C. A., Elmore, R. D., & Fuhlendorf, S. D. (2019). Behavioral modifications lead to disparate demographic consequences in two sympatric species. Ecology and evolution, 9(16), 9273-9289.
Line 75 = On the other hand..not at the other hand
Line 109: Describe always the acronym when you use it for the first time “Near Infrared” (NIR) and use the acronym only if you cite another time this word in the text.
Line 115 = Add the version of Google Earth used in this study
Line 124 – 146 = You add numerous and different acronyms in this part of the text…Do you use them in the other part of the ms? If yes, you can leave them in the ms but if you cite them only one time, I think that you can delete the acronym in the round brackets
Lines 188 – 189 (Figure 1) and 222 – 223 (Figure 2) = Can you distinguish even better the differences of colors gradations between cell colored reds and greens… if you just used different colors? (red, green, blu, yellow…etc…). What do you think?
Lines 232 – 284: This section of the ms seems to me very untreated. The authors should discuss their results highlighting both the strengths and the limits of their work. I think that this section should be extended including also comparisons with other methodologies used in the “land-cover classification”
Lines 286 – 293 = Call this section “Conclusions”
Author Response
Revision of the ms “Land-cover classification using MaxEnt: ¿Can we trust in model quality metrics for estimating classification accuracy?
Line 2 = delete the question mark upside down
Response: Question mark deleted as requested
Lines 10 – 11 = I prefer “…especially for species distribution modelling”
Response: Addressed in the new version
Lines 27 – 28: Place the keywords in alphabetic order
Response: We made the suggested changes
Line 34 = I prefer “…especially for species distribution modelling”
Response: We made the suggested change
Line 36 = After “…distribution modeling is MaxEnt”, you should add also the following reference:
Phillips, S. J., Anderson, R. P., Dudík, M., Schapire, R. E., & Blair, M. E. (2017). Opening the black box: An open‐source release of Maxent. Ecography, 40(7), 887-893.
Response: We believe that adding another reference to stress this point is unnecessary as we have included the original citation from Phillips et al 2008 (i.e. reference 4 in our manuscript). In addition, the reference suggested is more focused on the release of the source code and features of the new release than in the algorithm itself.
Line 43 = After “While MaxEnt was originally developed as a species distribution modeling software…”, you should add also the following recent references as examples:
Bosso, L., Ancillotto, L., Smeraldo, S., D’Arco, S., Migliozzi, A., Conti, P., & Russo, D. (2018). Loss of potential bat habitat following a severe wildfire: a model-based rapid assessment. International Journal of Wildland Fire, 27(11), 756-769.
Response: We made the suggested change
Line 47 = You have written two times “and California”.
Response: Addressed in the new version
Lines 71 – 72 = After “These quality metrics are applied to the probabilistic model generated by MaxEnt…”, you should add also the following recent references as examples:
Tanner, E. P., Orange, J. P., Davis, C. A., Elmore, R. D., & Fuhlendorf, S. D. (2019). Behavioral modifications lead to disparate demographic consequences in two sympatric species. Ecology and evolution, 9(16), 9273-9289.
Response: We added the reference as requested
Line 75 = On the other hand..not at the other hand
Response: Addressed in the new version
Line 109: Describe always the acronym when you use it for the first time “Near Infrared” (NIR) and use the acronym only if you cite another time this word in the text.
Response: We added the definition of NIR acronym in the new version as suggested.
Line 115 = Add the version of Google Earth used in this study
Response: Google Earth version was added as requested.
Line 124 – 146 = You add numerous and different acronyms in this part of the text…Do you use them in the other part of the ms? If yes, you can leave them in the ms but if you cite them only one time, I think that you can delete the acronym in the round brackets
Response: We used the acronyms in the figures and in the supplementary materials section.
Lines 188 – 189 (Figure 1) and 222 – 223 (Figure 2) = Can you distinguish even better the differences of colors gradations between cell colored reds and greens… if you just used different colors? (red, green, blu, yellow…etc…). What do you think?
Response: We explored different coloration schemes for these figures, but decided to use a red to green palette as these was considered the most easy to understand scheme, because we could use the white color to those relations not different from 0. In addition, we included the correlation values in each cell to provide additional information for readers.
The figure caption explicitly indicates the color differences “Squares are colored based on Rho values from green (+1) to red (-1)”
Response: See previous response.
Lines 232 – 284: This section of the ms seems to me very untreated. The authors should discuss their results highlighting both the strengths and the limits of their work. I think that this section should be extended including also comparisons with other methodologies used in the “land-cover classification”
Response: This article is a methodological and technical article that presents relevant information for user of MaxEnt as a one-class classification tool, but that have no guidelines on how to choose the “best model” for achieving the highest classification accuracy. In this regard, we did not evaluate how good is this tool in comparison with other methodologies as we consider this as an entirely different objective that is not addressed in this article. Furthermore, comparison of MaxEnt with other methods have already been addressed in previous articles (e.g. Mack and Waske, 2017, In-depth comparisons of MaxEnt, biased SVM and one-class SVM for one-class classification of remote sensing data. Remote Sensing Letters), which is referenced in our manuscript. Therefore, we consider that the article main message could lose relevance if we add to the discussion additional points that are not directly related with our objective. Nevertheless, we have included in the Supplementary Material all the original excel files with the classification accuracy metrics and models quality for all the models built, aiming to provide further information to readers for examining the levels of classification accuracy resulting for each parameter and threshold combinations.
Lines 286 – 293 = Call this section “Conclusions”
Response: The section “conclusions” is not compulsory for the journal hence the lack of the suggested section. However, we have placed the last paragraph as a new conclusion section.
Reviewer 3 Report
This manuscript presents a comparative analysis of different model quality metrics for the evaluation of classification uncertainty in an application of the Maximum Entropy approach to the analysis of land cover satellite images. The authors conclude that these quality metrics may be inappropriate and classification uncertainty is better estimated by measures such as overall accuracy or K coefficient.
This work is interesting for users of Maximum Entropy-based software and brings out sources of confusion in discriminating different land covers. I did not find particular methodological flaws but, in my opinion, presentation should be improved. As a general recommendation, I suggest a careful English editing.
Detailed commets:
Study area: It could be useful to present an image of the area (e.g., an orthophoto or an RGB composition of satellite data etc.) so that readers can see the mosaic of different land covers in the analyzed site.
Remote sensing imagery: I feel the assumption of 0% cloud cover is too ambitious. If the authors have used the Sentinel 2 masks, I would limit to say “cloud-free according to the Sentinel..”.
Metrics and Land cover classification: maybe, explicit formulas and/or discussions of the metrics as well as a detailed explanation of the one-class approach could be useful for non-expert readers.
Results: The authors present correlation estimates but absolute accuracy values are not discussed. I would be curious to learn about the accuracy we can obtain with MaxEnt in the examined case so that I could also roughly compare it with different approaches, maybe simpler.
Discussion: I think the first paragraph (lines 232-239) may be dropped.
Conclusion: Conclusions should be presented in a separate section.
Author Response
This manuscript presents a comparative analysis of different model quality metrics for the evaluation of classification uncertainty in an application of the Maximum Entropy approach to the analysis of land cover satellite images. The authors conclude that these quality metrics may be inappropriate and classification uncertainty is better estimated by measures such as overall accuracy or K coefficient.
This work is interesting for users of Maximum Entropy-based software and brings out sources of confusion in discriminating different land covers. I did not find particular methodological flaws but, in my opinion, presentation should be improved. As a general recommendation, I suggest a careful English editing.
Detailed comments:
Study area: It could be useful to present an image of the area (e.g., an orthophoto or an RGB composition of satellite data etc.) so that readers can see the mosaic of different land covers in the analyzed site.
Response: We have included in the Supplementary Material two composite Sentinel 2 satellite images for the area of study, specifically for the 16 x 16 km quadrant used for testing the classification accuracy, to show the heterogenous land covers and the phenological changes between winter and summer in the study area.
Remote sensing imagery: I feel the assumption of 0% cloud cover is too ambitious. If the authors have used the Sentinel 2 masks, I would limit to say “cloud-free according to the Sentinel..”.
Response: The study area is in a Mediterranean type of climate, procuring cloud free images in summer and winter is highly plausible (See Figure S1). In any case, we changed the “0% cloud cover” statement by “cloud-free images”
Metrics and Land cover classification: maybe, explicit formulas and/or discussions of the metrics as well as a detailed explanation of the one-class approach could be useful for non-expert readers.
Response: We conceived this study as a technical report that can provide new information to researchers and practitioners familiarized with remote sensing classification procedures, so we deemed no necessary to provide the formulas and explain the approach to readers. Furthermore, we provide the appropriate references in case readers need additional information on formulas and the one-class classification approach. Therefore, we consider t
Results: The authors present correlation estimates but absolute accuracy values are not discussed. I would be curious to learn about the accuracy we can obtain with MaxEnt in the examined case so that I could also roughly compare it with different approaches, maybe simpler.
Response: We have included in the Supplementary Material all the original excel files with the classification accuracy metrics and models quality for all the models built, aiming to provide further information to readers for examining the levels of classification accuracy resulting for each parameter and threshold combinations.
Discussion: I think the first paragraph (lines 232-239) may be dropped.
Response: We consider that this short paragraph is important to reconnect the reader to the main objective of the manuscript after two intricate technical sections.
Conclusion: Conclusions should be presented in a separate section.
Response: Conclusions section was added. This was also suggested by other reviewer.
Round 2
Reviewer 1 Report
I appreciate the answer of the authors and I indeed misunderstood certain parts of the manuscript, but this is because I believe the manuscript is somewhat confusing. I believe it is important you make a conceptual diagram that explains visually the methodological framework to obtain the LC. Also this statement ''To validate the classification results we used true-points obtained by visual interpretations of very-high resolution images available in Google Earth.'' implies some sort of subjectivity in the evaluation I believe. So the question is: why one would use MaxEnt vs other LC available products that are massively verified already? I believe a motivation for the use of MaxEnt is necessary. Is it better? Is is just another alternative? What is the impact?
I like the manuscript but I really believe you should explain more clearly the methods, the motivations for which one would use MaxEnt and the accuracy/sensitivity/complexity of MaxEnt output for LC classification vs already existing LC maps. Thx.
Author Response
Reviewer 1
I appreciate the answer of the authors and I indeed misunderstood certain parts of the manuscript, but this is because I believe the manuscript is somewhat confusing. I believe it is important you make a conceptual diagram that explains visually the methodological framework to obtain the LC. Also this statement ''To validate the classification results we used true-points obtained by visual interpretations of very-high resolution images available in Google Earth.'' implies some sort of subjectivity in the evaluation I believe. So the question is: why one would use MaxEnt vs other LC available products that are massively verified already? I believe a motivation for the use of MaxEnt is necessary. Is it better? Is is just another alternative? What is the impact?
I like the manuscript but I really believe you should explain more clearly the methods, the motivations for which one would use MaxEnt and the accuracy/sensitivity/complexity of MaxEnt output for LC classification vs already existing LC maps. Thx.
The objective of this article was not methodological but a technical article that could give some light in how obtain the best possible result using MaxEnt as land-cover classification tool. The use of this methodology is already described in another article already publish by the same authors. We did not think that it was relevant to add details of an already publish methodology.
About the points selected to validate the models, subjectivity is almost nonexistent due the nature of the classification used. It is almost impossible to confuse a tree with a building or grass with a tree.
About the advantages, the most common method is the Maximum Likelihood (ML) algorithm which is included in almost any GIS software available in the market. The problem with this method is that “assumes that pixel values from a given layer will be normally distributed”. Because of this it is “outperformed by newer non-parametric algorithms that do not require the assumption of an a-priori particular distribution”. There are several methods that solve to solve this issue such as Neural Network, Support Vector Machines and Maximum Entropy algorithms. However, MaxEnt is the only method that allows to classify one-class at a time avoiding the need to classify several classes so you can obtain one-class of interest. For example, if you are interested to determine the number of parks and the area present in a given city you just run one-class a not all the class present in a city such as infrastructure, rivers, etc. All these is already published in a previous article.
We hope that reviewer understand that these article was not meant to be a methodological article but a technical one where parametrization, model selection and classification accuracy was discussed which has been done for the niche modelling application of Maxent but have not done for the land classification arena.
Reviewer 2 Report
Perfect! All my questions were well addressed!
Author Response
Thank you very much for your help!
Reviewer 3 Report
Although I think this work meets the minimum requirements to be published in Entropy, in my opinion a very little effort could allow the paper to reach a broader audience. Providing essential formulas and explaining basic concepts, as I suggested in the first round, could be useful for readers which are searching for more efficient land cover classification methods but are not yet sufficiently prepared to understand the importance of a too cryptic work.
Author Response
Reviewer 3
Although I think this work meets the minimum requirements to be published in Entropy, in my opinion a very little effort could allow the paper to reach a broader audience. Providing essential formulas and explaining basic concepts, as I suggested in the first round, could be useful for readers which are searching for more efficient land cover classification methods but are not yet sufficiently prepared to understand the importance of a too cryptic work.
An article with the information that the reviewer mention is already publish by the same authors. There we give more in detail use of the methodology. This article is a follow up of the previous one and more technical in nature about how parametrization can affect the results. We recognize that the article can be classified as “dense” but we believe that we did our best effort to present the information in the clearest way possible.